# Effect of Le Fort I Maxillary Advancement and Clockwise Rotation on the Anteromedial Cheek Soft Tissue Change in Patients with Skeletal Class III Pattern and Midface Deficiency: A 3D Imaging-Based Prediction Study

**DOI:** 10.3390/jcm9010262

**Published:** 2020-01-18

**Authors:** Hsin-Chih Lai, Rafael Denadai, Cheng-Ting Ho, Hsiu-Hsia Lin, Lun-Jou Lo

**Affiliations:** 1Division of Orthodontics, Department of Dentistry, Chang Gung Memorial Hospital, Taoyuan 333, Taiwan; sam985tw@gmail.com (H.-C.L.); ma2589@gmail.com (C.-T.H.); 2Department of Plastic and Reconstructive Surgery and Craniofacial Research Center, Chang Gung Memorial Hospital, Chang Gung University, Taoyuan 333, Taiwan; denadai.rafael@hotmail.com; 3Image Lab and Craniofacial Research Center, Chang Gung Memorial Hospital, Taoyuan 333, Taiwan

**Keywords:** maxilla advancement, maxilla rotation, cheek mass, midface deficiency

## Abstract

Patients with a skeletal Class III deformity may present with a concave contour of the anteromedial cheek region. Le Fort I maxillary advancement and rotational movements correct the problem but information on the impact on the anteromedial cheek soft tissue change has been insufficient to date. This three-dimensional (3D) imaging-assisted study assessed the effect of surgical maxillary advancement and clockwise rotational movements on the anteromedial cheek soft tissue change. Two-week preoperative and 6-month postoperative cone-beam computed tomography scans were obtained from 48 consecutive patients who received 3D-guided two-jaw orthognathic surgery for the correction of Class III malocclusion associated with a midface deficiency and concave facial profile. Postoperative 3D facial bone and soft tissue models were superimposed on the corresponding preoperative models. The region of interest at the anteromedial cheek area was defined. The 3D cheek volumetric change (mm^3^; postoperative minus preoperative models) and the preoperative surface area (mm^2^) were computed to estimate the average sagittal movement (mm). The 3D cheek mass position from orthognathic surgery-treated patients was compared with published 3D normative data. Surgical maxillary advancement (all *p* < 0.001) and maxillary rotation (all *p* < 0.006) had a significant effect on the 3D anteromedial cheek soft tissue change. In total, 78.9%, 78.8%, and 78.8% of the variation in the cheek soft tissue sagittal movement was explained by the variation in the maxillary advancement and rotation movements for the right, left, and total cheek regions, respectively. The multiple linear regression models defined ratio values (relationship) between the 3D cheek soft tissue sagittal movement and maxillary bone advancement and rotational movements of 0.627 and 0.070, respectively. Maxillary advancements of 3–4 mm and >4 mm resulted in a 3D cheek mass position (1.91 ± 0.53 mm and 2.36 ± 0.72 mm, respectively) similar (all *p* > 0.05) to the 3D norm value (2.15 ± 1.2 mm). This study showed that both Le Fort I maxillary advancement and rotational movements affect the anteromedial cheek soft tissue change, with the maxillary advancement movement presenting a larger effect on the cheek soft tissue movement than the maxillary rotational movement. These findings can be applied in future multidisciplinary-based decision-making processes for planning and executing orthognathic surgery.

## 1. Introduction

The contour of the anteromedial cheek region has been considered as an important determinant of the overall aesthetics and youthfulness of the face [1,2,3]. For young and aesthetically pleasant faces, the position of the most convex point of the anteromedial cheek region, i.e., cheek mass position, is 1.6–2.1 mm in front of the cornea perpendicular line [1,4,5,6]. This cheek mass position has been considered as a standard metric for evaluating the anteromedial cheek contour [1,4,5], with the cheek point behind the cornea perpendicular line, i.e., a concave/deficient cheek contour, being considered a sign of ageing, exhaustion, and depression [4,5,7,8].

This deficiency of the anteromedial cheek contour has been corrected by using different approaches, ranging from nonsurgical (different injectable filler materials) to surgical procedures (fat grafting, facial implants, and mobilization of an osteotomized maxillary bone segment) [9,10,11,12,13]. The surgical maxillary mobilization, i.e., Le Fort I osteotomy, is a powerful therapeutic modality for a high number of patients who present with the skeletal Class III pattern associated with an anteromedial cheek deficiency and concave facial profile [12,13,14]. Instead of adopting isolated mandible setback surgery [15,16,17], the two-jaw orthognathic surgery, with the combination of maxillary advancement and clockwise rotation (occlusal plane alteration) movements, has become popular for the correction of patients with a Class III malocclusion, concave facial profile, and deficiency of the anteromedial cheek region [18,19]. However, quantitative data particularly related to the impact of both maxillary advancement and rotational movements on the three-dimensional (3D) midface soft tissue change have been insufficient to date.

A growing body of literature has appraised the relationship between facial bone and soft tissue changes after orthognathic surgery treatment, with most methodological designs being based on 3D landmark-based measurements [20,21,22,23,24,25]. To overcome the drawbacks of landmark-based techniques, a 3D facial region-based volumetric subtraction method was recently developed for the measurement of facial bone and soft tissue changes after orthognathic surgery [26]. This 3D imaging-assisted technique can demonstrate the average movement between corresponding bone and soft tissue models, generating accurate information for the prediction of the relationship between the bone and soft tissue change [26]. However, we are not aware of any study formally focused on the impact of 3D surgical maxillary bone advancement and clockwise rotational movements on the anteromedial cheek soft tissue change. Understanding this postoperative change may support multidisciplinary teams (including dentists, orthodontists, oral surgeons, maxillofacial surgeons, ear, nose, and throat surgeons, plastic surgeons, and head and neck surgeons) in surgical planning and execution, as well as to deliver appropriate preoperative counseling to patients and their parents and to set the preoperative expectations of the patients with respect to the midface component.

The primary purpose of this 3D computer-assisted study was to assess the effect of Le Fort I maxillary advancement and clockwise rotation movements on anteromedial cheek soft tissue change by using the region-based volumetric subtraction method [26]. We hypothesized that each type of maxillary bone movement would have a different effect on the 3D anteromedial cheek soft tissue change. The secondary purpose was to compare the 3D cheek mass position from orthognathic surgery-treated patients stratified by the level of the maxillary advancement movement to a 3D healthy normative data [6].

## 2. Methods

### 2.1. Study Population

This retrospective study recruited consecutive Taiwanese Chinese patients with a Class III malocclusion associated with an anteromedial cheek deficiency and concave facial profile who underwent orthodontic treatment and orthognathic surgery by the two senior authors (C.-T.H. and L.-J.L.) at the Chang Gung Craniofacial Center between July 2016 and July 2018. Demographic, clinical, surgical, and outcome (3D imaging-based data) data were collected. Patients were excluded from the study based on the following criteria: (1) additional maxillary osteotomies (i.e., surgically assisted maxillary expansion, anterior subapical osteotomy, segmented maxillary osteotomies, or the trimming of the anterior nasal spine/pyriform area), (2) cleft, syndromic diagnosis, or facial traumatic sequelae, (3) previous facial procedures, (4) any facial intervention performed within the observation period, (5) incomplete medical or imaging records, and (6) incomplete postoperative follow-up (of <6 months).

The included sample comprised 22 males and 26 females with a mean age of 23.1 years. In the clinical evaluation, all 48 patients had a concave facial profile, deficiency in the anteromedial cheek contour, and a protruding mandible. Radiographic examination revealed a Class III skeletal relationship, negative A point–nasion–B point angle, negative facial convexity angle, and negative overjet. Further pre- and post-operative characteristics of the included patients are provided in Table 1.

This study was performed with the approval of the Institutional Review Board (Chang Gung Medical Foundation = IRB201801386A3).

### 2.2. Orthognathic Surgery Treatment

Details of the preoperative and postoperative treatments adopted in this center have been previously described [26,27,28,29,30,31,32,33,34]. All included patients underwent the modified surgery-first approach with the 3D computer-aided single-splint two-jaw orthognathic surgery technique (Figure 1). A combination of different degrees of maxillary advancement and posterior impaction through Le Fort I osteotomy and mandibular setback through bilateral sagittal split osteotomy was adopted for the achievement of a normal dentoskeletal relationship. The decision of the degree of maxillary advancement and rotational movements was achieved by consensus between the orthodontic and surgeon professionals after the appraisal of the different elements, such as cephalometric parameters and facial proportion, balance, and symmetry. For patients with a skeletal Class III pattern, low occlusal plane, and brachycephalic facial type, the maxillary clockwise rotation (center of rotation at the maxillary incisors) was adopted to increase the midface fullness and the smile curve, decrease the maxillary incisors angulation, and achieve a balanced and aesthetic facial profile due to the consequent posterior chin rotation. The maxillary advancement also produced an increase in the midface fullness.

Three-dimensional printed surgical wafers and 3D maxillary measurements (medial and lateral maxillary pillars bilaterally) were adopted to transfer the virtual planning to the actual surgery. After the achievement of the desired bone repositioning, the Le Fort I was initially fixed by using 2 mm titanium miniplates and 6 mm screws placed on the nasomaxillary and zygomatico-maxillary pillars bilaterally. The proximal ramus segment was then placed in a relaxed position, gently pushed up to ensure the position of the condylar head in the glenoid fossa, and fixed by the percutaneous insertion of three bicortical screws 14–16 mm long. No interpositional bone graft was used. Intermaxillary fixation was released and the occlusion was evaluated. Genioplasty was finally executed as planned, along with intraoperative judgement. The patients with no intermaxillary fixation were admitted into the regular ward for two days following the surgery and then were clinically examined based on regular surgical and orthodontic appointments. A liquid diet was recommended in the first week, followed by a soft diet in the second week. On average, a 1–3-month period of presurgical orthodontic treatment was performed for arch form compatibility. No postoperative intermaxillary fixation was performed, and the orthodontic therapy was started 2–4 weeks postoperatively. The routine removal of miniplates and screws is not adopted in this center.

### 2.3. Three-Dimensional Image Acquisition and Processing

Preoperative (2 weeks before surgery) and postoperative (6 months after surgery) 3D maxillofacial images were acquired using an i-CAT^TM^ cone-beam computed tomography (CBCT) scanner (Imaging Sciences International, Hatfield, PA, USA). The extended field of view (FOV) was 22 (height) × 16 (depth) cm, the scanning time was 40 s, and the voxel size was 0.4 × 0.4 × 0.4 mm. The patient’s head was positioned with the Frankfort horizontal (FH) plane parallel to the ground. Throughout the scan, the patients were instructed not to swallow, keep their mouth closed, and maintain a centric occlusion bite. The 6-month postoperative time was adopted for this study to avoid the influence of the swelling process on the soft tissue analysis, as previously demonstrated [35,36].

The image data were stored in Digital Imaging and Communications in Medicine (DICOM) format and processed with a slice thickness of 0.4 mm. The 3D bone and soft tissue facial models were processed and analyzed using the SimPlant O and O software (Materialize Dental, Leuven, Belgium). To assess the preoperative and postoperative maxillary bone and midface soft tissue structures, the postoperative 3D model was superimposed on the corresponding preoperative 3D model at the skull base, forehead, and orbital areas (non-operated parts) by using the best-fit method [26,33]. Accuracy of superimposition was checked by the root-mean-square deviation tool (3dMD Vultus software, 3dMD LLC, Atlanta, GA, USA), with a value of ≤0.5 mm certifying precision [26,30,33,34] (Figure 2). Both the facial bone and the soft-tissue components were concomitantly addressed by alternating the type of imaging modality and 3D models in the SimPlant software.

### 2.4. Three-Dimensional Anatomical Landmarks, Planes, and Measurement Parameters

Soft tissue and bone-derived 3D anatomical landmarks, reference planes, and measurement (i.e., anteromedial check soft tissue sagittal movement, surgical maxillary advancement, surgical maxillary rotation, and cheek mass position) parameters (Figure 3; Table 2) were defined based on previous studies [6,26,27,28,29,30,33].

### 2.5. Three-Dimensional Facial Region of Interest and Calculations

To evaluate the soft tissue changes associated with the underlying bone movements, the anteromedial cheek volumetric change caused by the orthognathic surgery was measured. The region of interest was selected based on previous findings that movement within each region was uniform [26]. These were defined by the intersection of four reference planes to represent the bilateral anteromedial cheek regions (Figure 4). By adopting the superimposed preoperative and postoperative 3D facial models, the bone and soft tissue volumetric change (mm^3^) models were automatically created by using the subtraction tool: postoperative minus preoperative models (Figure 5).

The preoperative 3D anteromedial cheek region models and the 3D bone and soft tissue volumetric change models were exported as STL (stereolithography) files (Figure 5) for further evaluation using Magics 13.0 software (Materialize Dental, Leuven, Belgium). The value of the preoperative surface area (mm^2^) and volumetric change (mm^3^) for each anteromedial cheek region were automatically computed. The average sagittal movement (mm) of soft or bone tissues in each anteromedial cheek region was obtained as the volumetric change (mm^3^) divided by preoperative surface area (mm^2^) [26].

All included patients were uniformly distributed based on maxillary advancement incremental values. The maxillary bone sagittal movement (mm) was stratified in four levels from the minimum to maximum values: 1–2 mm, 2–3 mm, 3–4 mm, and >4 mm. The amount of maxillary rotational movement (degree) was also calculated: postoperative maxillary rotational angle minus preoperative maxillary rotational angle. Both the maxillary bone sagittal (advancement) movement (mm) and maxillary rotational movement (degree) parameters were adopted for assessment of the relationship between the bone and soft tissue (anteromedial cheek soft tissue sagittal movement) change.

### 2.6. Measuring the Cheek Mass Position

Postoperative 3D facial soft tissue models were adopted for the calculation of the postoperative cheek mass position, i.e., the most prominent point of the anteromedial cheek contour [6], using SimPlant software (Figure 6). The previously published 3D facial soft-tissue normative data, i.e., the cheek mass position (2.14 ± 1.20 mm), derived from the healthy Taiwanese Chinese population (*n* = 60; 30 male and 30 female individuals with normal occlusion and balanced facial profile) were adopted for comparative analysis [6].

### 2.7. Reliability

Twenty 3D scans were randomly selected for reliability tests. After the orientation of the 3D digital models at a standardized position, all anatomical landmark identifications were performed twice by two independent evaluators blinded to the type of the surgical maxillary movement, with a 2-week interval between each measurement. The Euclidean distance between the first and second landmark coordinates was calculated.

### 2.8. Statistical Analysis

In the descriptive analysis, the mean was used for metric variables, and percentages were given for categorical variables. It was verified that the data were normally distributed by using the Kolmogorov–Smirnov test. An independent samples *t*-test was adopted for statistical comparisons between patients and healthy individuals. An intraclass correlation coefficients (ICCs) test was adopted for intra-evaluator and inter-evaluator reliability analysis. The ICC for absolute agreement based on a two-way random effects analysis of variance (ANOVA) was calculated. Scatter plots were generated to investigate the relationship between the soft tissue changes and the underlying bone tissue movements, and a Pearson correlation test was calculated for each corresponding pair [37,38,39]. The relationships between two variables were considered weak, moderate, strong, or extremely strong when their Pearson correlation coefficients (absolute value of *r*) were inferior to 0.3, between 0.3 and 0.6, between 0.6 and 0.8, and larger than 0.8, respectively. Multiple linear regression analyses were computed to evaluate the relationship between the bone and soft tissue movements, resulting in coefficients of determination (*R*^2^) [37,38,39]. *R*^2^ values represent the proportion of the variance in the dependent variable (anteromedial cheek soft tissue sagittal movement) that is predictable from the independent variables (maxillary bone advancement and rotational movements). Predictive regression models (equations) were also developed for the total cheek region and the left and right cheek regions. A power analysis (G*Power 3.1.6; Heinrich-Heine-Universität Düsseldorf, Düsseldorf, Germany) conducted for the regression analysis parameter indicated that at least 48 patients were required for an α of 0.05 and a power of 80%. Two-sided values of *p* < 0.05 were considered statistically significant. All analyses were performed using SPSS Version 18.0 (IBM Corp., Armonk, NY, USA).

## 3. Results

### 3.1. Accuracy and Reliability

The mean root-mean-square deviation value was 0.42 mm, ranging from 0.35 to 0.48 mm for all 3D superimposed models, ensuring the accuracy of the registration. The intra-evaluator and inter-evaluator reliability were considered excellent for all measurements, with ICCs values of 0.945 (0.903–0.997) and 0.916 (0.843–0.979), respectively.

### 3.2. Soft Tissue and Bone Movements

Comparisons between the right and left sides revealed no significant difference (all *p* > 0.05) for anteromedial cheek soft tissue sagittal movement, maxillary bone sagittal movement, and maxillary bone rotational movement (Table 3).

### 3.3. Ratio of 3D Soft Tissue to Bone Sagittal Movement

The appraisal of the ratio of the 3D soft tissue to 3D bone sagittal movement (Table 4) revealed that the anteromedial cheek soft tissue movement followed the underlying bone movement.

### 3.4. Effect of the Maxillary Advancement and Rotational Movements

No significant interaction (all *p* > 0.05) was observed between the two types of maxillary bone movements (advancement versus rotation) for the right, left, and total cheek regions.

Scatter plots revealed extremely strong positive linear relationships (all *r* > 0.8; all *p* < 0.01) between the anteromedial cheek soft tissue sagittal movement and maxillary bone advancement and rotational movements (Figure 7).

Multiple linear regression analyses revealed a significant effect of the maxillary advancement (all *p* ≤ 0.001) and maxillary rotation (all *p* ≤ 0.006) on the anteromedial cheek soft tissue sagittal movement. Overall, 79% of the variation in the anteromedial cheek soft tissue sagittal movement was explained by the variation in the surgical maxillary bone advancement and rotation movements. All predictive regression models revealed a larger effect of the maxillary advancement than the maxillary rotational movement. Overall, the relationships for maxillary bone advancement/cheek soft tissue and maxillary bone rotation/cheek soft tissue were of the ratios 1:0.63 and 1:0.07, respectively (Table 5).

The patients who had maxillary advancements of 1–2 mm or <3 mm presented a significantly larger average 3D cheek mass position (all *p* < 0.001) value compared to the patients who had maxillary advancements of 3–4 mm or >4 mm, with no significant difference (all *p* > 0.05) in comparisons between 1–2 mm versus <3 mm and 3–4 mm versus >4 mm (Table 6).

Maxillary advancements of 1–2 mm or <3 mm had a significantly inferior (all *p* < 0.05) average 3D cheek mass position value compared to the 3D healthy Taiwanese Chinese normative cohort. Maxillary advancements of 3–4 mm and >4 mm had no significant difference (all *p* > 0.05) when compared to the 3D healthy norm value (Table 6).

## 4. Discussion

Patient satisfaction for the correction of facial sagittal deformity and malocclusion via orthognathic surgery requires not only the accomplishment of a functional occlusion status but also the achievement of facial symmetry, balance, proportion, and aesthetics [32,40,41]. Patients are more concerned with improving their facial appearance than with the underlying skeletofacial framework change [32,40,41]. Therefore, a reliable prediction of the postoperative facial soft tissue change is of paramount importance when appraising orthognathic surgery-related outcomes. Many methods have been adopted for measuring soft tissue responses to underlying skeletal changes, with selected anatomical landmark-based techniques being the most common tools adopted [20,21,22,23,24,25]. However, for example, the closest point color-coded maps method does not allow for the measurement of homologous or corresponding distances. Moreover, most prior studies have focused on the impact of a specific type of surgical maxillary mobilization, i.e., maxillary advancement, on the facial soft tissue change [20,21,22,23,24,25,42,43,44], with the maxillary rotation receiving less attention to date. It is consequently fundamental that further outcome analyses are conducted by implementing a well-delineated 3D imaging-guided methodology based on an accurate quantitative evaluation of facial bone and soft tissue changes after both maxillary advancement and rotational movements.

This study contributes to the literature by providing the effect not only of maxillary advancement but also maxillary rotational movements on the anteromedial cheek soft tissue change measured by the ratio, correlation, and regression parameters. Instead of adopting the point-to-point movement’s method, this study employed an advanced biomedical engineering software-based 3D CBCT-derived methodology using a validated, reliable, and accurate 3D system package. This imaging technique enables the combination of the 3D bone and soft-tissue imaging modalities, as required for the present study design. Instead of using the widely applied anatomical landmarks-guided linear and angular measurements of distance between two 3D objects, the employed 3D computer-assisted analyses take advantage of all facial surface-related data points characterizing a more global and precise evaluation of the region of interest. Previously defined facial morphometric parameters were applied to define the 3D regions of interest. All the threshold segmentation values used for CBCT-based 3D model reconstructions were chosen by an experienced and trained researcher, and the measurement process was identified twice by two different evaluators. The difference in the bone and soft tissue between the pre- and post-orthognathic surgery treatment was measured by creating the 3D soft tissue models and corresponding bone models for all patients. The region of interest, i.e., the anteromedial cheek area, was precisely defined by the intersection of bone- and soft-tissue-derived reference planes, because the planes itself were constructed on fixed, unchanging points, which did not have a direct influence on the surgery. All the anatomical identifications were performed on the 3D models with concomitant checking in each CBCT-derived slice. To attenuate the influence of other facial areas (the lateral supracommissural, upper lip, and nose regions), which also present with orthognathic surgery-induced change, were not considered for analysis. Moreover, superimposed 3D facial models facilitated the consideration of the actual bone and soft tissue change for investigation. The accuracy of the superimposition and the excellent reliability scores revealed that the current virtual-guided data were consistently collected.

Overall, the scatter plots and multiple linear regression analyses demonstrated that the anteromedial cheek soft tissue sagittal movement can be accurately predicted based on the type of maxillary bone movement. The appraisal of the scatter plots displayed extremely strong (all *r* > 0.8) positive linear relationships between anteromedial cheek soft tissue sagittal movement and maxillary bone advancement and rotational movements. The results revealed high multiple correlation coefficients of determination (*R*^2^ values between 0.788 and 0.789), indicating that the dependent variable (cheek soft tissue movement) had a significant direct dependence on the variation of the independent variables (maxillary bone advancement and rotational movements). In total, 78.9%, 78.8%, and 78.8% of the variations in the anteromedial cheek soft tissue movement were explained by the variation in the maxillary advancement and rotational movements for the right, left, and total cheek regions, respectively. Moreover, all of the multiple linear regression models could be used to predict the cheek soft tissue movement from the corresponding bone movements, there existing a larger effect of the maxillary advancement on the cheek soft tissue sagittal movement than the maxillary rotational movement. For example, in the total cheek region, a cheek soft tissue advancement of 0.627 mm could be expected with a maxillary advancement of 1 mm, and a cheek soft tissue advancement of 0.070 mm could be expected with a maxillary rotation of 1 degree.

Prior orthognathic surgery-related investigations have revealed a soft tissue to bone sagittal movement ratio of 0.63–0.68, 0.74, and 0.79 by employing different 3D imaging modalities (CBCT, laser scan, and optical surface scan, respectively) [21,26,44]. The overall trend found in our study, i.e., a positive correlation between bone and soft tissue changes, as well the total soft tissue to bone sagittal movement ratio value (0.73), is in agreement with these published studies [21,26,44]. However, our findings may not exactly be similar to previously reported values [21,26,44] because of the differences in sample compositions, population bio-characteristics, and 3D methodological approaches.

In this study, we also showed that the postoperative cheek mass position was altered according to the level of the maxillary advancement movement. We demonstrated that maxillary advancements larger than 3 mm were required to achieve an average cheek mass position similar to a 3D healthy normative value [6]. Maxillary advancements smaller than 3 mm had significantly lower cheek mass position value compared to 3D healthy normative value, which may clinically represent a flat to concave contour of the anteromedial cheek region.

Potential caveats of this study should be addressed. We assessed the anteromedial cheek soft tissue region-related change and position because it is a relevant clinical parameter for the planning and execution of Le Fort I maxillary osteotomy and mobilization in the treatment of patients with a Class III malocclusion associated with a midface deficiency and concave facial profile. However, the current results cannot be extrapolated to other facial regions, requiring further investigation.

Moreover, as no significant interaction was observed between the maxillary advancement and maxillary rotation parameters, the effect of each type of bone movement could be independently analyzed. Future studies testing different hypotheses can apply a different methodology to appraise the relationship between the types of bone movements.

As our findings are restricted to a specific group of young adult patients with a skeletal Class III pattern associated with a deficiency of the anteromedial cheek contour who were managed by two-jaw orthognathic surgery procedures, any generalizations must be undertaken with caution. The adoption of a standardized approach for surgical planning and execution, as previously described by our group [27,28,29,30,31,32,33,34], partially limited the bias of orthodontic-, surgical technique-, and professional-related factors while interpreting our results. Future investigations may address the impact of different planning and execution techniques on the anteromedial cheek soft tissue change. Patient-perceived outcome related to the anteromedial cheek region change should also be the target of a further evaluation.

Despite these drawbacks, the current findings could support future orthognathic surgery-based research and clinical practice. As the selection of a proper outcome measurement tool is a key component that influences the value of outcome-based research, the methodology adopted in this study, i.e., 3D imaging-guided bone and soft tissue measurements and regression analysis, can be applied and adapted to test other facial bone surgery-related hypotheses. For clinical practice, multidisciplinary teams can provide patient-centered care, including the establishment of patient-specific planning of orthognathic surgery treatment (Figure 8), the supporting of computer-based prediction programs by employing the predictive regression models, and delivering better education for future patients and parents based on the 3D anteromedial cheek soft tissue change versus maxillary advancement and rotational movements and 3D cheek mass position versus level of the maxillary advancement.

## 5. Conclusions

This 3D imaging-assisted study demonstrates that: (1) both Le Fort I maxillary advancement and rotational movements affected the anteromedial cheek soft tissue change, (2) the maxillary advancement had a larger effect on the anteromedial cheek soft tissue sagittal movement than the maxillary rotational movement, and (3) maxillary advancements larger than 3 mm resulted in a postoperative cheek mass position similar to healthy norms.

## Figures and Tables

**Figure 1 jcm-09-00262-f001:**
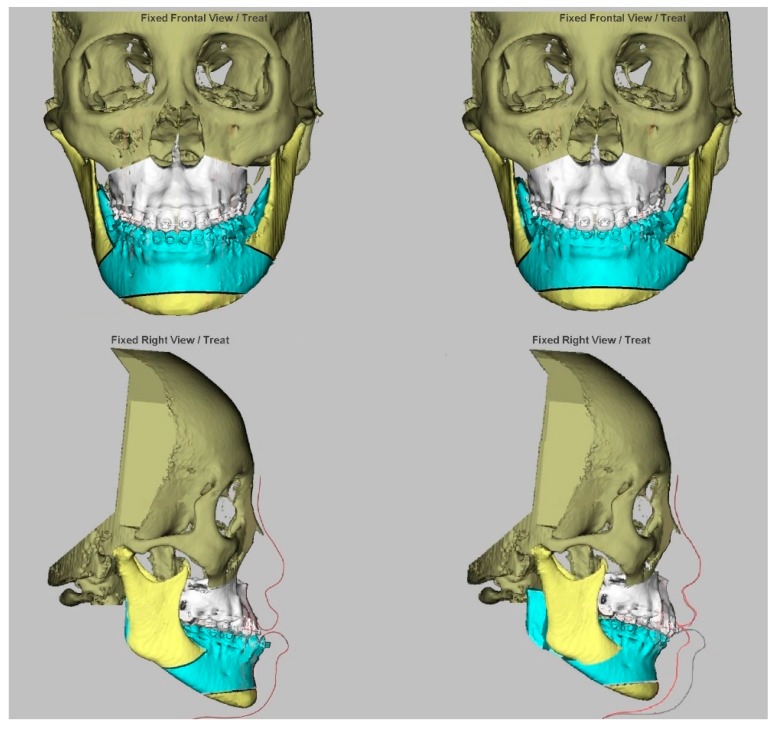
Virtual imaging-assisted simulation of single-splint two-jaw orthognathic surgery technique (final occlusal splint, 1-piece Le Fort 1 maxillary osteotomy, and bilateral sagittal split osteotomy) for correction of skeletal Class III deformity. (**Right**) Before (**left**) and after mobilization of the maxillomandibular complex, including translational (advancement) and rotational (pitch clockwise) to achieve a symmetric, balanced, and proportional face. The maxillomandibular complex rotations in the yaw and roll directions were also adopted for final skeletal positioning and adjustments, such as the elimination of residual asymmetry of contour and interference between the osteotomized bony structures and regions at the pterygoid maxillary junction and mandibular ramus areas.

**Figure 2 jcm-09-00262-f002:**
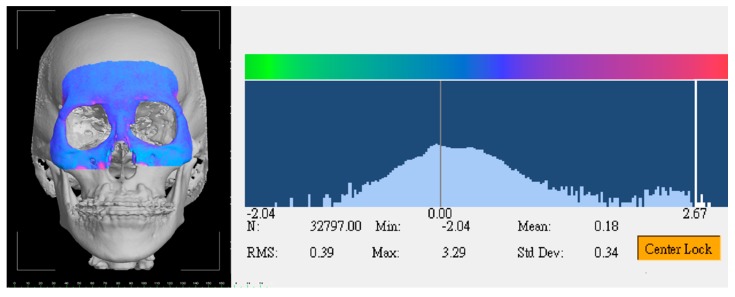
The root-mean-square (RMS) deviation tool for accuracy examination between preoperative and postoperative three-dimensional (3D) facial models at non-operated forehead and orbital regions.

**Figure 3 jcm-09-00262-f003:**
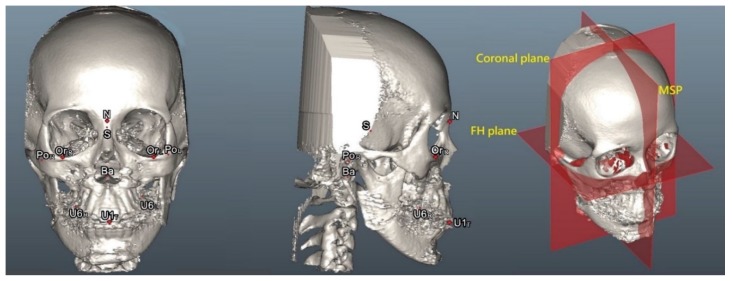
Three-dimensional skeletofacial model displaying the anatomical landmarks and reference planes, including the horizontal plane (X axis; Frankfort horizontal plane, FH), the sagittal plane (Y axis; midsagittal plane, MSP), and the coronal plane (Z axis). For definitions, please refer to Table 2.

**Figure 4 jcm-09-00262-f004:**
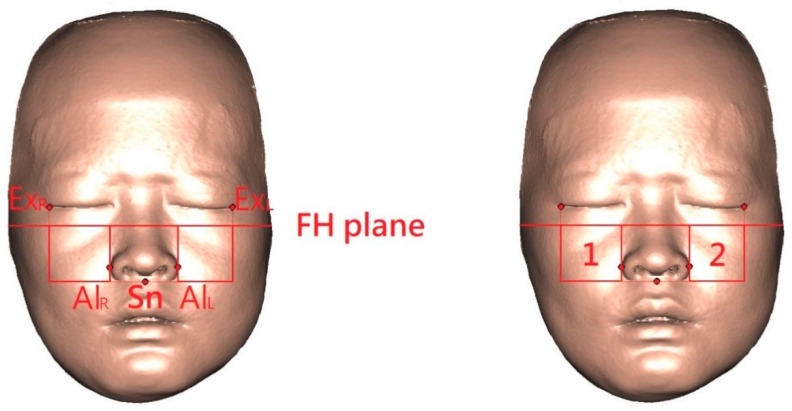
(**Left**) Defining the 3D regions of interest by four interconnected bone- and soft-tissue-derived reference planes. Al, Alare; Sn, subnasal; FH, Frankfort horizontal plane; Ex, Exocanthion; R, right side; L, left side. (**Right**) 3D cheek soft tissue models were constructed in the right and left anteromedial cheek regions (squares 1 and 2, respectively). Bilateral 3D cheek bone models corresponding to the soft tissue models were also constructed (see Figure 5).

**Figure 5 jcm-09-00262-f005:**
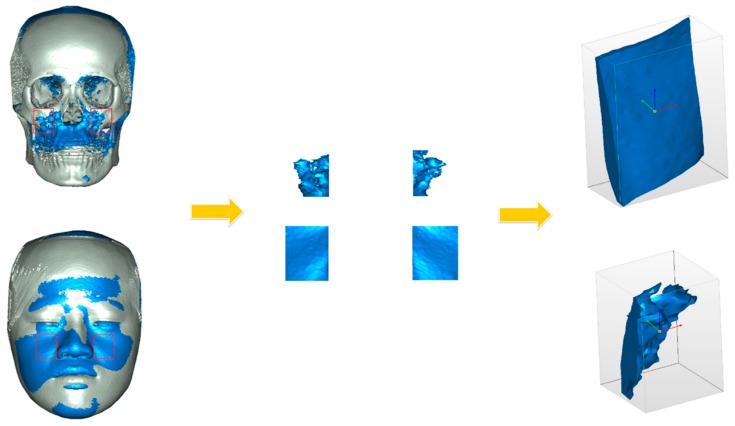
Workflow from registration to calculation of 3D bone and soft tissue volumetric change in the anteromedial cheek region, bilaterally. (**Left**) Superimposed preoperative and postoperative 3D facial bone and soft tissue models. (**Center**) The computer-generated 3D overall volumetric changes (mm^3^) were processed as postoperative minus preoperative 3D models. (**Right**) 3D bone and soft tissue models representing the overall volumetric change were exported to Magics 13.0 software for further calculation.

**Figure 6 jcm-09-00262-f006:**
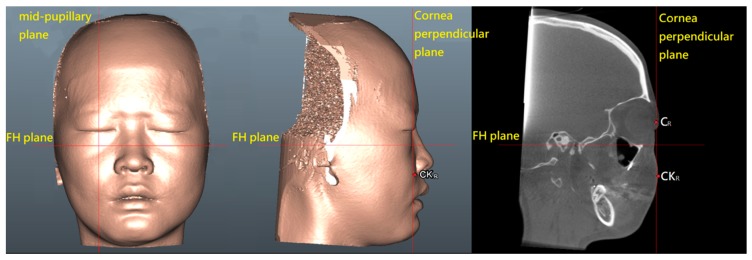
Defining the cheek mass (CK) position for the cheek prominence measurement. To represent the most projected point on the anteromedial cheek contour, the CK point was located on the most convex point on the mid-pupillary plane (vertical line) below the infraorbital area. The horizontal distance between CK point and mid-pupillary plane was measured. C, cornea point; FH, Frankfort horizontal plane; R, right side. For definitions, please, refer to Table 2.

**Figure 7 jcm-09-00262-f007:**
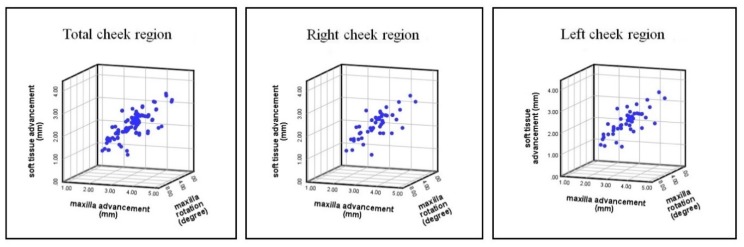
Scatter plots of correlation between the anteromedial cheek soft tissue region movement (mm) and the Le Fort I maxillary advancement (mm) and rotational (degrees) movements, with *r* = 0.892, *r* = 0.889, and *r* = 0.890 for the right, left, and total cheek regions (all *p* < 0.01).

**Figure 8 jcm-09-00262-f008:**
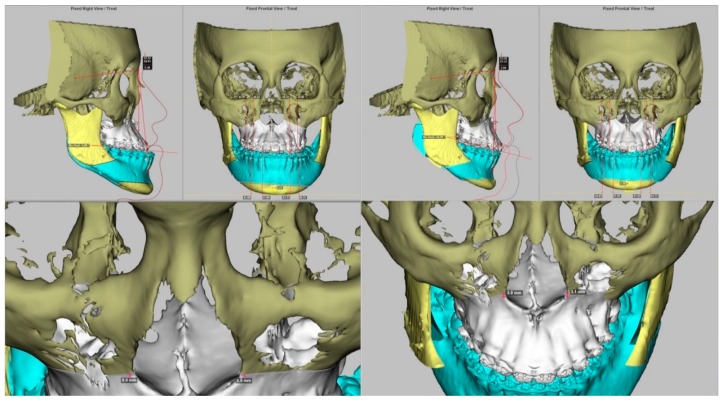
Practical examples of different compositions of the maxillary advancement and rotational movements for two-jaw orthognathic surgery in patients with a skeletal Class III pattern and midface deficiency. (**Left**) Actual deformity. (**Right**) Surgical simulation. The decision regarding the magnitude of each movement is based on key elements, including an accurate preoperative clinical evaluation- and 3D image-based diagnosis of the actual skeletofacial deformity (bone and soft tissue elements), the definition of patients’ complaints and requests, the achievement of a balanced, symmetric, and aesthetically pleasant facial appearance during computer-assisted simulation, and the surgical feasibility of the simulated bone mobilizations.

**Table 1 jcm-09-00262-t001:** Pre- and post-operative characteristics of included patients.

Parameters	Pre-Orthognathic Surgery		Post-Orthognathic Surgery		*p*-Value
	Mean	SD	Mean	SD	
SNA (°)	83.776	4.149	86.887	4.224	<0.001
SNB (°)	87.787	4.379	84.231	4.006	<0.001
ANB (°)	−4.412	3.048	2.953	2.435	<0.001
A-N vertical (mm)	−2.052	2.351	0.978	1.694	<0.001
Occlusal plane (°)	8.348	4.507	11.102	5.236	<0.001
Mandibular plane-FH plane angle (°)	28.151	5.460	26.793	4.158	0.049
Angle convexity (°)	−9.466	5.118	2.641	3.401	<0.001
Facial convexity (°)	−3.222	6.717	8.429	4.803	<0.001
Sn-G’ vertical (mm)	4.363	1.101	6.031	1.254	<0.001
Nasolabial angle (°)	95.607	13.146	109.111	10.697	<0.001

° degree; mm millimeters; SD: standard deviation; SNA sella–nasion–A point; SNB sella–nasion–B point; ANB A point–nasion–B point; FH Frankfort horizontal plane.

**Table 2 jcm-09-00262-t002:** Definition of the three-dimensional anatomical landmarks, reference planes, and measurement parameters.

Parameters	Abbreviations	Definitions
Soft tissue landmarks		
Exocanthion	Ex_L_, Ex_R_	The most lateral points at the outer commissure of the eye fissure
Cornea	C_L_, C_R_	The most anterior point of the cornea
Alare	Al_L_, Al_R_	The most lateral point on each alar contour
Subnasale	Sn	The midpoint of the angle at the columella base where the lower border of the nasal septum and the surface of the upper lip meet
Check mass	Ck_L_, Ck_R_	The most anterior point on the mid-pupillary plane (MPP) under infraorbital area and ahead of cornea perpendicular plane (CPP)
Bone landmarks		
Orbitale	Or_L_, Or_R_	The most inferior point of the infra-orbital rim
Porion	Po_L_, Po_R_	The most superior point of the external acoustic meatus
Nasion	N	The junction between the nasal and frontonasal sutures
Sella turcica	S	The center point of the sella turcica
Basion	Ba	The most anterior point of the foramen magnum
Dental landmarks		
U1 incisal tip	U1	The midpoint between the crowns of the maxillary central incisors tip
U6 cusp	U6_L_, U6_R_	The most inferior point of the mesial cusp of the crown of the first upper molar in the profile plane
Reference planes		
Frankfort horizontal plane	FH	The plane passes the bilateral porion points and the midpoint between the bilateral inferior orbital points
Midsagittal plane	MSP	The plane perpendicular to the FH plane and passing through the sella turcica (S) and the nasion (N) points
Coronal plane	CO	The plane perpendicular to the FH and MSP planes and passing through the sella turcica (S) point
Occlusal plane	OP	The plane passes the midpoint between upper incisor tips and the midpoint between upper molar buccal cusps
Mid-pupillary plane	MPP	The plane perpendicular to the FH and CO planes and passing through the cornea (C) point
Cornea perpendicular plane	CPP	The plane perpendicular to the FH and MSP planes and passing through the cornea (C) point
Lateral cheek plane	LCP	The plane perpendicular to the FH plane and passing through the exocanthion (Ex) point
Medial cheek plane	MCP	The plane perpendicular to the FH plane and passing through the alare (Al) point
Inferior cheek plane	ICP	The plane parallel to the FH plane and passing through the subnasale (Sn) point
Angle measurement		
Maxillary rotational angle	-	The angle formed between the FH and OP planes

L left side; R right side.

**Table 3 jcm-09-00262-t003:** Three-dimensional soft tissue and bone movements.

Parameters	Right Side	Left Side	Total
	(m ± SD)	(m ± SD)	(m ± SD)
Cheek soft tissue sagittal movement (mm)	2.180 ± 0.695	2.173 ± 0.691	2.176 ± 0.689
Maxillary bone sagittal movement (mm)	2.971 ± 0.826	2.969 ± 0.828	2.970 ± 0.823
Maxillary bone rotation (°)	3.341 ± 2.346	3.341 ± 2.346	3.341 ± 2.346

° degree; mm millimeters; m: mean; SD: standard deviation.

**Table 4 jcm-09-00262-t004:** Ratio of 3D soft tissue to bone sagittal movement.

Regions	Soft Tissue/Bone Ratio (m ± SD)
Right side	0.733 ± 0.128
Left side	0.731 ± 0.125
Total	0.732 ± 0.126

m: mean; SD: standard deviation.

**Table 5 jcm-09-00262-t005:** Multiple linear regression analyses and models of the relationships between the anteromedial cheek soft tissue movements and maxillary advancement and rotational movements.

Regions of Interest	Predictive Regression Models	Coefficient of Determination (*R*^2^) *	Partial R-squared (X_1_, X_2_)	Maxillary Advancement (*p*-Value)	Maxillary Rotation (*p*-Value)
Right cheek region	*Y* = 0.625*X*_1_ + 0.073*X*_2_	0.789	0.423, 0.046	<0.001	0.003
Left cheek region	*Y* = 0.629*X*_1_ + 0.067*X*_2_	0.788	0.436, 0.040	<0.001	0.006
Total cheek region	*Y* = 0.627*X*_1_ + 0.070*X*_2_	0.788	0.432, 0.043	<0.001	<0.001

*Y* anteromedial cheek soft tissue sagittal movement (mm); *X*_1_ maxillary advancement (mm); *X*_2_ maxillary rotation (degree); * *R*^2^ < 0.5 is meaningful; *R*^2^ = 0.79 may be interpreted as follows: 79% of the variance in the response (dependent) variable can be explained by the explanatory (independent) variables, with the remaining 21% being attributed to unknown factors. Postoperative check mass position.

**Table 6 jcm-09-00262-t006:** Comparison of the check mass position between orthognathic surgery-treated patients and 3D healthy Taiwanese Chinese normative data.

Level of the Maxillary Advancement (*n* = 96, 100%)	Check Mass Position (mm)			
	Post-Orthognathic Surgery	Healthy Taiwanese Chinese Norm	Difference *	*p*-Value
1–2 mm (n = 14, 14.6%)	1.307 ± 0.621	2.145 ± 1.201	−0.837 ± 0.621	0.018
2–3 mm (n = 34, 35.4%)	1.491 ± 0.607	2.145 ± 1.201	−0.654 ± 0.607	0.004
3–4 mm (n = 34, 35.4%)	1.906 ± 0.531	2.145 ± 1.201	−0.238 ± 0.531	0.191
>4 mm (n = 14, 14.6%)	2.375 ± 0.723	2.145 ± 1.201	0.230 ± 0.723	0.498

n: number of cheek sides; mm: millimeters; * post-orthognathic surgery value minus healthy Taiwanese Chinese normative value.

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
