# Peer review of "Effect of Le Fort I Maxillary Advancement and Clockwise Rotation on the Anteromedial Cheek Soft Tissue Change in Patients with Skeletal Class III Pattern and Midface Deficiency: A 3D Imaging-Based Prediction Study"

_jcm, 2020, doi:10.3390/jcm9010262_

Round 1

Reviewer 1 Report

This study investigates the the impact of maxillary advancement and mandibular clockwise rotation on cheek mass position in patients with Class III skeletal pattern and mid face deficiency. They find that both maxillary advancement and rotation have a significant effect on cheek mass position and that greater levels of maxillary advancement restore a normal cheek mass position whereas lower levels do not. The results are perhaps not particularly surprising but mostly this is a well-presented, well-conducted study. I have only some minor comments.

ln 117 'appraisal of the multiparameter'. This phrase does not make sense to me. What is the multiparameter?

ln 184-187 'Based on a prior definition...anteromodal cheek regions'. This sentence is strange and clunky, I would suggest re-writing it : "the region of interest was selected based on previous findings that movement within each region was uniform. these were defined by the intersection of four ... etc"

ln 210 'linearly distributed'. It's not clear to me what this means. Do you mean uniformly distributed? normally distributed?

ln 240 'paired t-test'. I think this must be wrong and you mean an independent groups t-test.

ln 240 please state the type of ICC used. there are many different types and they have very different meanings. 

ln 247 'between the average bone and soft tissue movements'. the analysis is not conducted on the averages as far as I can tell.I believe the correct phrasing here is 'between the bone and soft tissue movements'.

Table 5 consider reporting also the partial R-squared, measuring the effect sizes of each of maxillary advancement and maxillary rotation individually

I felt that some of 'limitations' paragraphs of the discussion were quite weak, and the authors are dreaming up limitations for the sake of it:

paragraph ln 379-387 I don't think the fact that your study is novel is a limitation of it.

paragraph ln 387-394 your results do not suggest the need to consider gender as an independent variable. Of course it might be necessary to do so in other studies...but this is obvious. Gender is a confound that most researchers will consider as a matter of course.  I think this discussion is not really relevant to the current study at all.

Reviewer 2 Report

I enjoyed reading this article. It has always been an area of interest on the change that happens in the mid-facial region. This paper provides some insight to just orthognathic surgery and future work should determine if it is adequate alone. I look forward to future work.
